# A tissue-like platform for studying engineered quiescent human T-cells' interactions with dendritic cells

Enas Abu-Shah[1,2]*, Philippos Demetriou[1†], Štefan Bálint[1†], Viveka Mayya[1], Mikhail A Kutuzov[2], Omer Dushek[2]*, Michael L Dustin[1,3]*

[1]Kennedy Institute of Rheumatology, University of Oxford, Oxford, United Kingdom; [2]Sir William Dunn School of Pathology, University of Oxford, Oxford, United Kingdom; [3]Skirball Institute of Biomolecular Medicine, New York University School of Medicine, New York, United States

**Abstract** Research in the field of human immunology is restricted by the lack of a system that reconstitutes the *in-situ* activation dynamics of quiescent human antigen-specific T-cells interacting with dendritic cells. Here we report a tissue-like system that recapitulates the dynamics of engineered primary human immune cell. Our approach facilitates real-time single-cell manipulations, tracking of interactions and functional responses complemented by population-based measurements of cytokines, activation status and proliferation. As a proof of concept, we recapitulate immunological phenomenon such as CD4 T-cells' help to CD8 T-cells through enhanced maturation of DCs and the effect of PD-1 checkpoint blockades. In addition, we characterise unique dynamics of T-cell/DC interactions as a function of antigen affinity.

**\*For correspondence:**
enas.abushah@path.ox.ac.uk (EA-S);
omer.dushek@path.ox.ac.uk (OD);
michael.dustin@kennedy.ox.ac.uk (MLD)

[†]These authors contributed equally to this work

## Introduction

The field of human immunology has gained much attention in the last few decades with the increasing realisation that despite the tremendous utility of mouse models, they can only provide limited insight for advancing translational research (*Wagar et al., 2018*; *Council MRC, 2008*). Most of the research in human immunology is based on observational studies, whereby, for example, data sets of gene expression profiles are used to identify disease biomarkers. Although these studies promote our understanding and help in developing hypotheses about disease mechanisms, they fall short, and experimental research is needed (*Davis, 2008*). The classical tools in experimental human immunology rely heavily on immortalised cell lines, stimulated clonal lines, peripheral blood, and more rarely humanised mouse models. Each of these tools has its limitation. Cell lines have diverged tremendously from primary cells, making their signalling machinery and response thresholds no longer representative (*Colin-York et al., 2019*); bulk peripheral blood assays lack the spatial organisation of tissues; and humanised mouse models have only a portion of their system 'humanised', rendering the remaining parts confounding factors (*Theocharides and Manz, 2018*). To address these limitations, we introduce a holistic approach to allow researchers to manipulate and study human immune cells in a tissue-like environment with minimal aberrations to their natural behaviours. We report a toolkit that achieves two critical goals for engineering T-cells: (1) high efficiency TCR expression and maintenance of T-cell motility, activation dynamics and viability using mRNA electroporation and (2) a tissue-like 3D culture (*Figure 1*).

**eLife digest** The human immune system protects the body from infection or cancer by detecting foreign and abnormal elements, known as antigens, and initiating a response to clear them. It relies on a type of white blood cell called a T-cell to distinguish which substances are the body's own, and which are infectious. Each T-cell is designed to attack a specific antigen, which they recognise using a unique 'T-cell receptor'. During an infection, immune cells called 'antigen presenting cells' hold out antigens for the T-cells to look at. Only when an antigen matches the T-cell's receptor can the T-cell get activated and trigger an immune response.

There are still gaps in our understanding about how human T-cells interact with antigen presenting cells. Since only a small number of T-cells in the human blood have the same receptor, it is difficult to collect the large number of identical T-cells needed to study this interaction. In addition, it is impractical to image how these interactions occur in a living human body.

Now, Abu-Shah et al. have developed a new system that engineers human T-cells to have the same specific receptor. T-cells collected from human blood received the genetic information for identical receptors via a technique called electroporation. This involves mixing the cells with a single-strand copy of the receptor gene and then applying electric pulses to make the cell membranes leaky so the code for the receptor can get inside the cells.

To study the interaction between these genetically engineered T-cells and antigen presenting cells, Abu-Shah et al. created a three-dimensional system that mimics the environment T-cells normally experience inside the body. T-cells cultured using this system behaved similarly to immune cells in the human body, and displayed the characteristics needed to trigger an immune response.

With this new system, researchers could recreate other aspects of the human immune response outside of the body, incorporating different types of immune cells and different genetic modifications. Not only could this improve our understanding of the human immune system, it could also be used as a way to screen specific drugs during pre-clinical studies.

## Results

### Engineering quiescent primary T-cells

Successful engineering of T-cell specificity requires the stable expression of a functional exogenous TCR, as well as the preservation of important cellular characteristics. One such characteristic is cell motility, which is essential for efficient antigen search in a physiological 3D setting. Here, we have optimised the procedures to engineer naïve human CD4 and CD8 T-cells using mRNA electroporation. The common approach to render T-cells antigen-specific relies on lentiviral transduction of the TCR, which requires prior expansion of T-cells to achieve high expression efficiency and hinders the study of natural quiescent populations, namely naïve and memory T-cells (*Dai et al., 2009*). To avoid the need for cell expansion, it is possible to use electroporation to deliver the genetic materials. However, reported approaches using electroporation suffer from low efficiency of expression, low viability, and most importantly impaired cell motility, a critical consideration in the context of delivery to target tissues. Using a square-wave electroporation apparatus, we are able to express the **1G4** TCR, specific to the NY-ESO tumour associated antigen (*Chen et al., 2000*) in naïve (*Figure 2A*) and memory (*Figure 2—figure supplement 1A*) CD8 T-cells as well as the **868** TCR specific for the HIV gag protein (*Varela-Rohena et al., 2008*) (*Figure 2—figure supplement 1B*), both presented by HLA-A*0201. Successful pairing of the alpha and beta heterodimer of the TCR is important for the expression and assembly at the cell surface, this was achieved by introducing cysteine modifications in the TCR chains (*Figure 2—figure supplement 1B* and *Table 1*) (*Cohen et al., 2007*), as well as co-transfection with the human CD3ζ (*Figure 2—figure supplement 1C*), as the endogenous protein is most likely sequestered by the endogenous TCR limiting the expression of the introduced TCR. The functionality of naïve CD8 T-cells following TCR electroporation was confirmed by the formation of a mature immunological synapse in naïve and activated CD8 T-cells (IS, *Figure 2B* and *Figure 2—figure supplement 1D–E*), and the upregulation of the activation marker CD69 (*Figure 2—figure supplement 2A*). The efficiency of TCR expression varied among donors with an average value of 81 ± 7% (mean ± SD, n = 13), with more than 90% cell viability and 80–90%

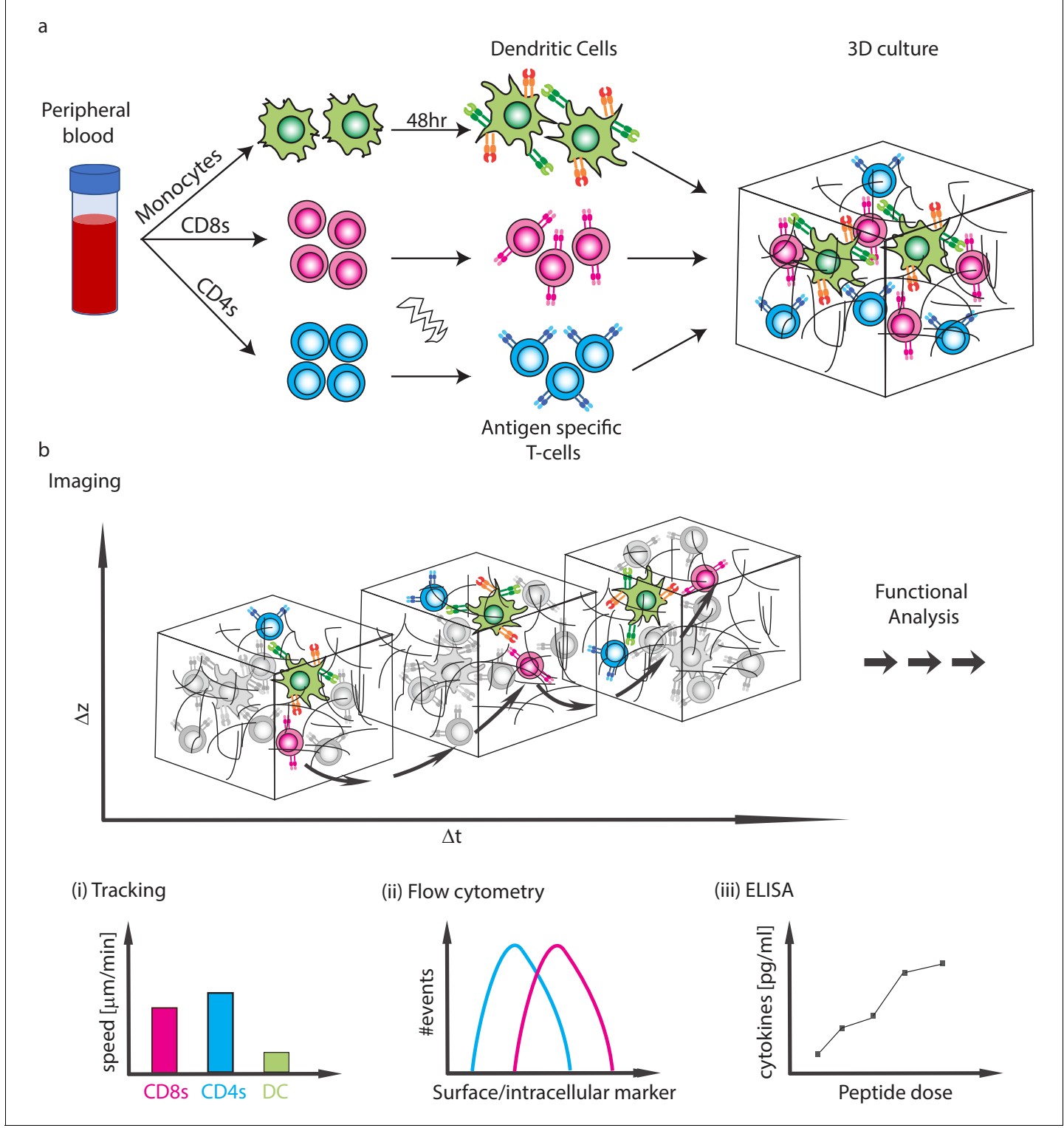

**Figure 1.** Schematics of the experimental system. (**a**) Isolation of CD4 T-cells, CD8 T-cells and monocytes from peripheral blood. Subpopulations of T-cells can be further purified and TCR expression is induced by mRNA electroporation. Monocytes are differentiated into DCs and are activated with a 48 hr express protocol. The cells are then moved into collagen gel-based 3D culture. (**b**) Cells can be taken for imaging or downstream functional analyses. (**i**) The imaging can be used to analyse interaction dynamics such as speed or conjugate formation, (**ii**) Cells can be extracted for flow cytometry, (**iii**) Cytokine secretion into the culture can be measured.

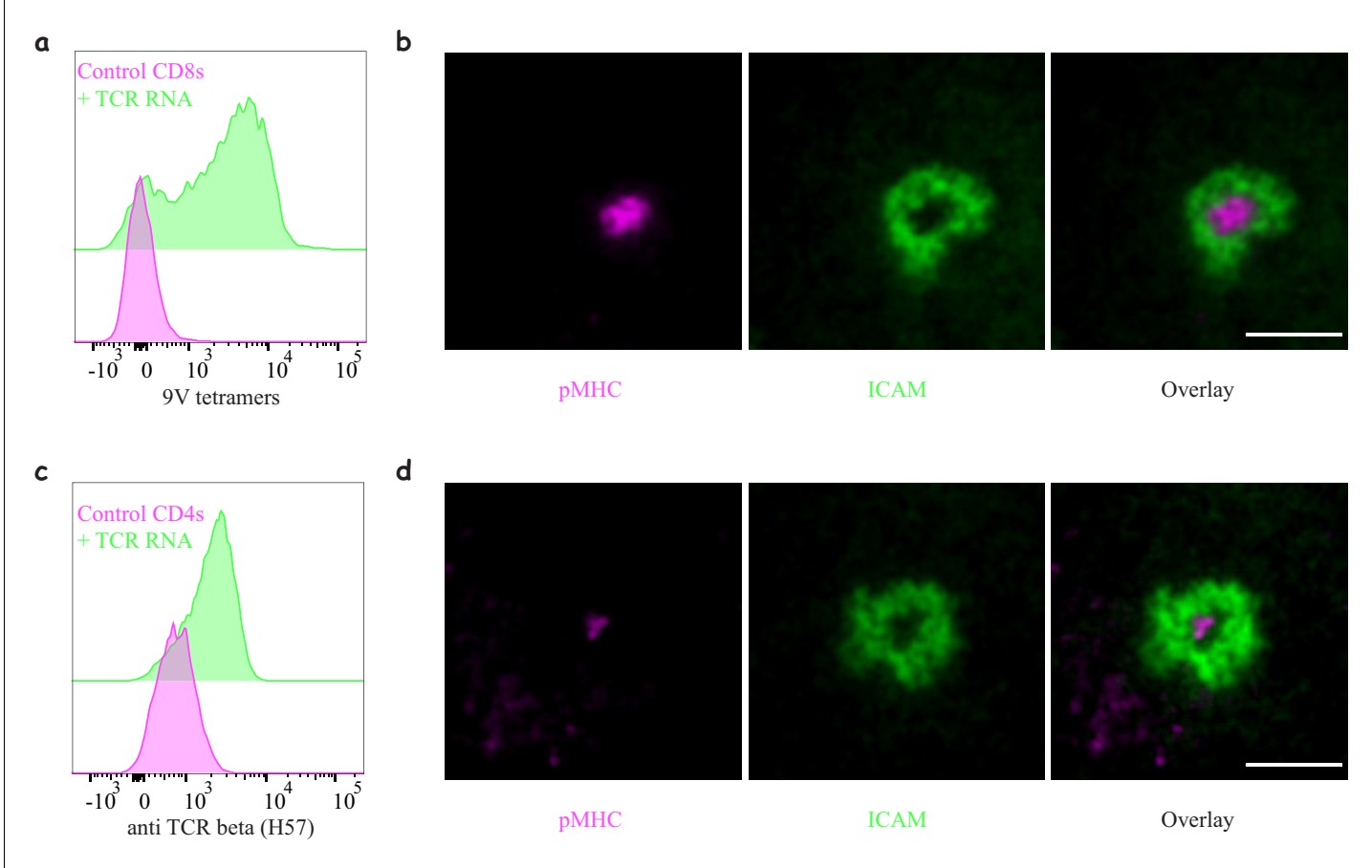

**Figure 2.** Engineering antigen-specific T-cells. (**a**) Expression of **1G4** TCR in naïve CD8 T-cells upon mRNA electroporation detected using NY-ESO-9V/HLA-A2 tetramer,~80% positive (Representative of N = 13). (**b**) Formation of an immunological synapse by **1G4**-expressing naïve CD8 T-cells on supported lipid bilayers (SLBs) with cSMAC enriched with NY-ESO-9V/A2 pMHC (magenta) surrounded by an LFA1/ICAM1 ring (green). Representative of >3 independent repeats. (**c**) Expression of **6F9** TCR in naïve CD4 T-cells detected using an antibody against the constant region of mouse TCRβ,~67% positive (Representative of N = 15). (**d**) Formation of an immunological synapse by **6F9**-expressing naïve CD4 T-cells on SLB containing MAGE/DP4 pMHC (magenta). Representative of >3 independent repeats. Scale bars = 5 µm.

The online version of this article includes the following source data and figure supplement(s) for figure 2:

**Source data 1.** TCR expression following mRNA electroporation.
**Figure supplement 1.** Engineering CD8 T-cells with different TCR constructs.
**Figure supplement 2.** Functional response of mRNA electroporated T-cells.
**Figure supplement 3.** Efficiency of electroporation and cell recovery using different methods.
**Figure supplement 4.** Engineering CD4 T-cells with different TCR constructs.
**Figure supplement 5.** Interactions of **1G4**-expressing naïve CD8 T-cells with pMHC presented on spatially segregated stimulatory spots.

cell recovery, both of which were severely reduced using alternative electroporation approaches (*Figure 2—figure supplement 3*).

The induction of expression of an exogenous TCR in CD4 T-cells is considerably harder than in CD8 T-cells (*Dai et al., 2009*). Using natural sequences or the introduction of cysteine modifications used for **1G4** and **868** failed to induce expression of TCRs in naïve CD4 T-cells. However, replacing

**Table 1.** Sequence modification of MHC-I restricted TCRs.

| TCR chain | Original sequence | Modified sequence |
|---|---|---|
| Alpha | DKTVL | DKCVL |
| Beta | GVSTD | GVCTD |

**Table 2.** Sequence alignment of human and mouse constant region used to modify MHC-II restricted TCRs

| TCR chain | Sequence alignment | | |
|---|---|---|---|
| Alpha | Human | IQNPDPAVYQLRDSKSSDKSVCLFTDFDSQTNVSQSKDSDVYITDKTVLDMRSMDFKSNS | 60 |
| | Mouse | IQNPEPAVYQLKDPRSQDSTLCLFTDFDSQINVPKTMESGTFITDKTVLDMKAMDSKSNG | 60 |
| | | ****:******:*.:*.*.::*\*\*\*\*\*\*\*\*\* **.:: :*...:********::** ***. | |
| | Human | AVAWSNKSDFACANAFNNSIIPEDTFFPSPESSCDVKLVEKSFETDTNLNFQNLS<u>VIGFR</u> | 120 |
| | Mouse | AIAWSNQTSFTCQDIFK----ETNATYPSSDVPCDATLTEKSFETDMNLNFQNLS<u>VMGLR</u> | 116 |
| | | *:****::.*:* : *:        :: :**.: .**..*.\*\*\*\*\*\*\* *********:*:* | |
| | Human | <u>ILLLKVAGFNLLMTLRLW</u>SS | 140 |
| | Mouse | <u>ILLLKVAGFNLLMTLRLW</u>SS | 136 |
| | | \*\*\*\*\*\*\*\*\*\*\*\*\*\*\*\*\*\* | |
| Beta | Human | DLNKVFPPEVAVFEPSEAEISHTQKATLVCLATGFFPDHVELSWWVNGKEVHSGVSTDPQ | 60 |
| | Mouse | DLRNVTPPKVSLFEPSKAEIANKQKATLVCLARGFFPDHVELSWWVNGKEVHSGVSTDPQ | 60 |
| | | **.:* **:*::****:***::.********* ************************** | |
| | Human | PLKEQPALNDSRYCLSSRLRVSATFWQNPRNHFRCQVQFYGLSENDEWTQDRAKPVTQIV | 120 |
| | Mouse | AYKE----SNYSYCLSSRLRVSATFWHNPRNHFRCQVQFHGLSEEDKWPEGSPKPVTQNI | 116 |
| | | . **     .:  *************:***********:****:*:*.:. .***** : | |
| | Human | SAEAWGRADCGFTSVSYQQGVLSATILYE<u>ILLGKATLYAVLVSALVLMAM</u>VKRKDF | 176 |
| | Mouse | SAEAWGRADCGITSASYQQGVLSATILYE<u>ILLGKATLYAVLVST</u>LVVMAMVKRKNS | 172 |
| | | \*\*\*\*\*\*\*\*\*\*\*:**.\*\*\*\*\*\*\*\*\*\*\*\*\*\*\*\*\*\*\*\*\*:**:\*\*\*\*\*\*\*: | |

Polar Non-polar Positively charged Negatively charged
Transmembrane segments are underlined. The alignments were generated using ClustalW

the constant region of the human TCR with that of mouse TCR (*Table 2*) (*Cohen et al., 2006*) allowed for successful chain pairing and we were able to achieve expression in 72 ± 8.9% (mean ± SD, n = 15) of the cells detected using H57 mAb against the mouse constant region of the β chain. We have successful expressed three different HLA-DPB1*04 restricted TCRs in naïve CD4s: **6F9** (*Figure 2C*) and **R12C9**, both specific to the MAGE-A3 protein (*Yao et al., 2016*) and **SG6** specific for NY-ESO (*Zhao et al., 2006a*) (*Figure 2—figure supplement 4A*). The expression of **SG6** was consistently lower than **6F9** and **R12C9** in naïve CD4 T-cells. However, when the same TCRs were electroporated into recently activated CD4 T-cells, high expression levels were achieved in 97.8 ± 1% (mean ± SD, n = 15) of the cells (*Figure 2—figure supplement 4B*). The functionality of the naïve CD4 T-cells upon TCR electroporation was confirmed by IS formation (*Figure 2D* and *Figure 2—figure supplement 4C–D*) and T-cell activation (*Figure 2—figure supplement 2B*). The enhanced expression of the TCR in activated CD4 T-cells resulted in enhanced pMHC accumulation at the IS (*Figure 2—figure supplement 4D*).

Using the square-wave electroporator we were able to maintain the motile behaviour and interaction dynamics of naïve CD8 T-cells expressing **1G4**, at similar levels to those of untouched cells, on 2D stimulatory 'spots' (*Figure 2—figure supplement 5*, *Video 1* and *Mayya et al., 2018*).

## "Express" monocyte-derived dendritic cells as model antigen presenting cells

Naïve T-cells get activated by interacting with professional antigen presenting cells (APCs); the most potent are dendritic cells (DCs). The closest model for *bona fide* DCs is monocyte-derived DCs (moDCs) with well-established protocols for their generation. The caveat with those protocols is their extended duration (7–10 days) (*Zhou and Tedder, 1995*), a time-frame which may alter autologous T-cells' phenotype. We therefore optimised a 48 hr protocol (*Obermaier et al., 2003*) to generate fully matured moDCs. During the first 24 hr monocyte differentiation into DC was induced, followed by 24 hr maturation to get activated DCs (acDCs) (*Figure 3*). The 'express' moDCs are able to upregulate MHC and other costimulatory molecules and produce a similar profile of soluble factors as the 'classical' moDC (*Figure 3—figure supplement 1A–B*). We have also quantified peptide loading

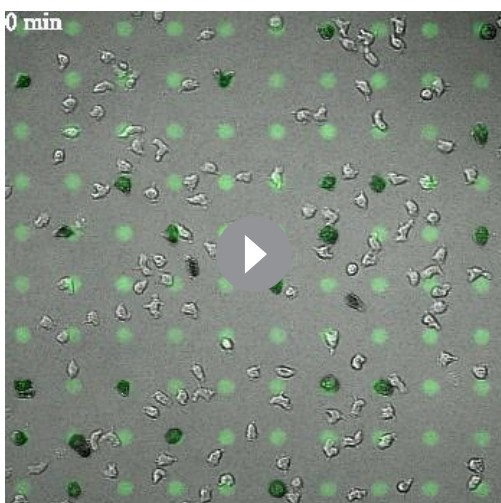

**Video 1.** A time-lapse movie showing a gradual increase in the number of arrested cells on stimulatory spots presenting pMHC after some transient interaction events. The image series is a composite overlay of DIC, Interference reflection (IRM) and fluorescence images of the cells and microcontact-printed biotinylated-FcIg (green) where cells form an IRM signal indicating spreading and durable interaction with the pMHC presented on stimulatory spots. Scale = Each circular spot is 10 µm in diameter. https://elifesciences.org/articles/48221#video1

on these cells and achieved levels similar to those previously reported with other APCs (*Figure 3—figure supplement 1C*) (*Zehn et al., 2006*).

## Collagen-based 3D model to support immune cell migration and interactions

T-cells–DC interactions have been extensively studied in mice using explants and intravital imaging (*Miller et al., 2002*); however, such studies are practically impossible in humans. Furthermore, accurate control and manipulation of parameters such as antigen dose and cell ratios is limited. Having successfully engineered naïve human T-cells we set out to establish a flexible 3D platform to interrogate their dynamics and interactions with APCs using a multitude of correlated functional readouts. The desired platform should: (1) support the culture of T-cells and APCs for multiple days, (2) support the motility of the cells, (3) allow live imaging, (4) enable downstream analysis and (5) be easy to use. To achieve these goals we chose collagen I -based 3D matrices (*Gunzer et al., 2000*), which we optimised to support the culture of human immune cells. Culturing cells in the presence of human serum results in better basal motility than FBS (*Figure 4A*, *Figure 4—figure supplement 1A* and *Video 2*, *Table 3*), which is significantly enhanced by the addition of homeo-

static chemokines such as CCL19 (*Video 3*), CCL21 and CXCL12 (*Video 4*). However, only CXCL12 was able to maintain high motility in cultures with FBS. We have explored different sources of collagen I (undigested and trypsin digested bovine and human collagen) and other complex extracellular matrices, ECM (*Figure 4—figure supplement 1B*). All collagens were equally good in supporting motility and interactions, while more advanced ECM may have a marginal advantage for motility. Yet, the larger batch variation, higher background activation and the complexity in extracting cells for downstream analysis weighed against adopting them for our assays. In our view, the use of bovine collagen-I as the 3D scaffold, with human serum, and CXCL12 to enhance motility provides an optimal 3D system with similar movement parameters to T-cells in mouse lymph nodes or explanted human tissues (*Bougherara et al., 2015*; *Murooka et al., 2012*; *Salmon et al., 2012*; *Woolf et al., 2007*). We have also re-evaluated two additional electroporation methods, namely Lonza-Amaxa and ThermoFisher-Neon. In addition to their lower performance in cell recovery and protein expression (*Figure 2—figure supplement 3*), they further resulted in poor motility dynamics and hence activation of cells within the 3D environment, where the search for antigen loaded APCs becomes a confounding factor (*Figure 4—figure supplement 1C–F*).

## Low affinity peptides result in T-cell activation without stable immunological synapse

We present here a proof of concept of the usefulness of our set-up by interrogating naïve CD8 T-cells activation. We used live cell imaging to follow the interactions in real-time by upregulation of CD69 (*Figure 4B* and *Video 5*) as well as calcium flux (*Figure 4C* and *Video 6*). We observed major changes in interaction dynamics as a function of peptide affinity (*Aleksic et al., 2010*) (*Figure 4D*). A high affinity peptide, NY-ESO-9V, (*Video 7*) induced T-cell arrest, whereas a lower affinity peptide, NY-ESO-9L, (*Video 8*), led to similar dynamics non-peptide loaded (*Video 9*) where no stable contacts are formed. Since our 3D culture offers a relatively simple way to extract cells for flow cytometry and downstream analyses, we were able to correlate the interaction dynamics with the activation

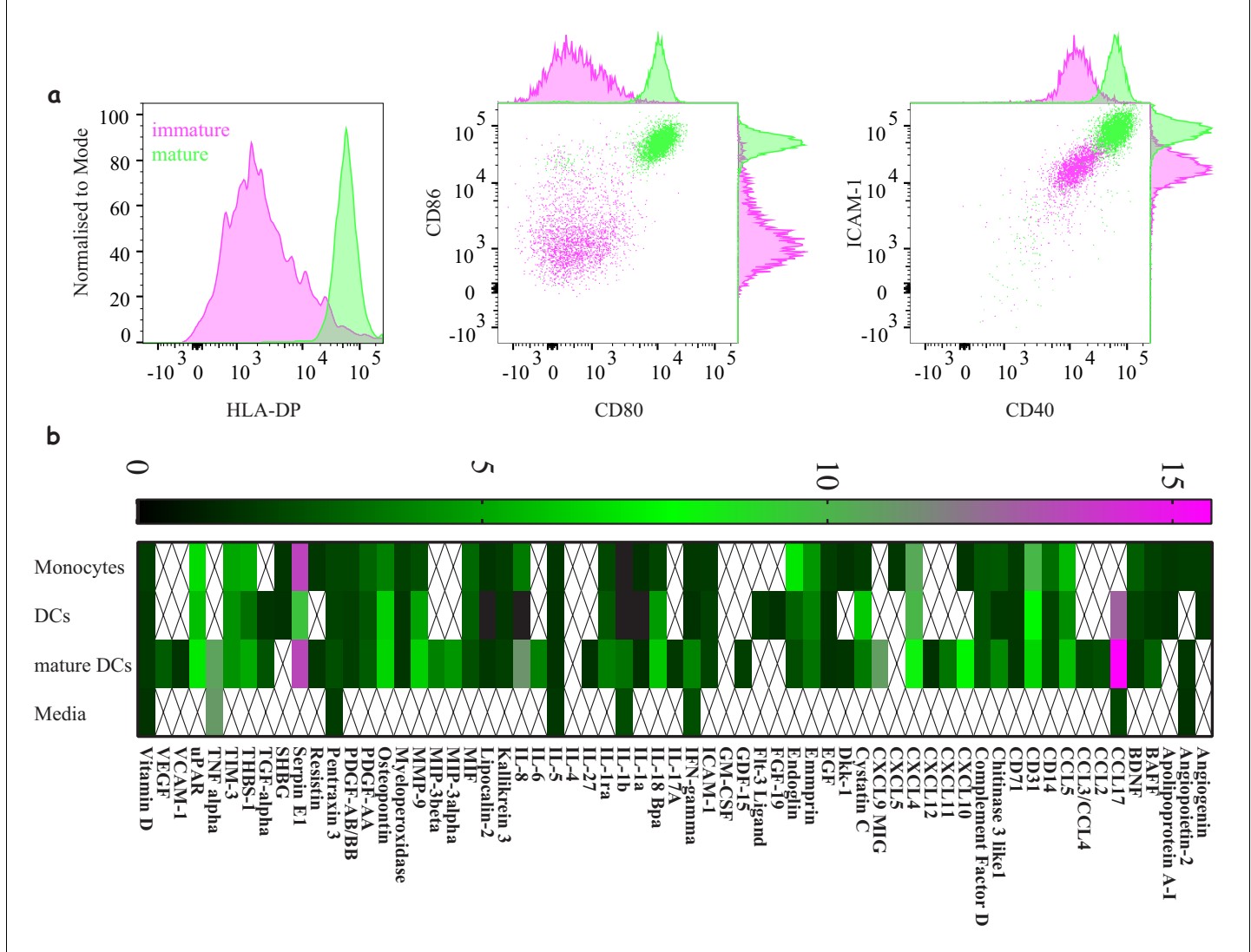

**Figure 3.** Characterising 'express' dendritic cells as a model antigen presenting cell. (a) Activation and differentiation profile of 'express' monocyte-derived dendritic cells: mature cells (green) upregulate their antigen presentation and costimulatory molecules compared to immature cells (magenta). Representative of >3 independent repeats. (b) Cytokine and chemokine secretion profile from monocytes, dendritic cells and mature dendritic cells using the 48 hr express protocol. Average values for three donors where signals bellow 1.5-fold above background were not included.
The online version of this article includes the following source data and figure supplement(s) for figure 3:

**Source data 1.** Cytokine production by classical DC.
**Figure supplement 1.** Comparing 'express' and 'classical' monocyte-derived dendritic cells.

state of the cells and observe marginal differences in the response relative to the major differences in dynamics (*Figure 4E*, *Figure 4—figure supplement 1A*). Using our system, we were also able to monitor cytokine production (*Figure 4—figure supplement 1B*), proliferation (*Figure 4—figure supplement 1C*) and intracellular markers (*Figure 4—figure supplement 1D*). In order to study the interaction dynamics of T-cells at different time points we could pre-load the DCs prior to or following gel polymerisation to mimic the delivery of antigen to lymph nodes (*Figure 4—figure supplement 1E* and *Videos 10–11*, respectively).

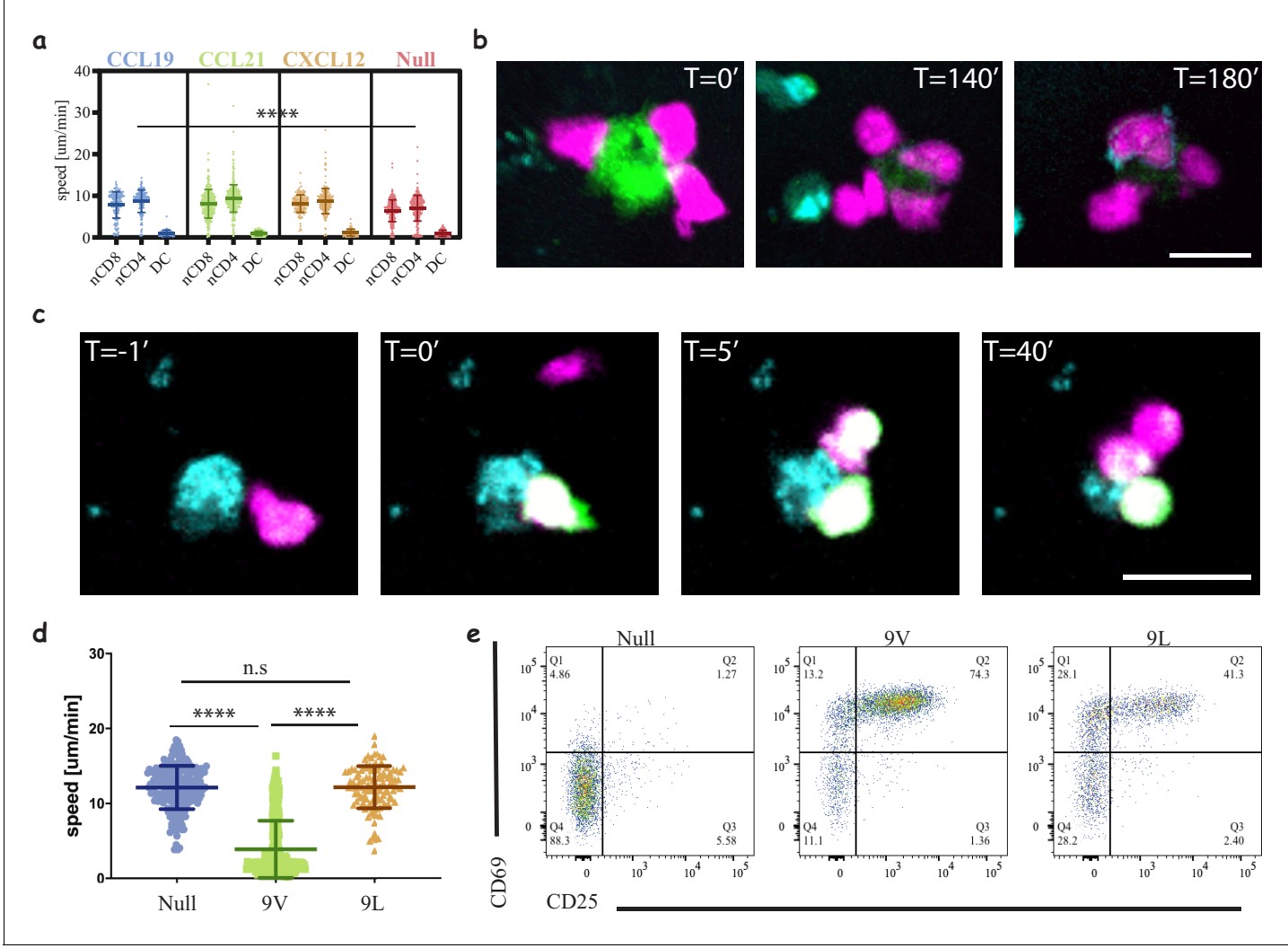

**Figure 4.** Three-dimensional culture system to study immune cell interactions. (a) Motility speeds of naïve CD4 and CD8 T-cells and acDCs in collagen gels with different chemokines. Note the faster motility of T-cells compared to acDCs and similar motility using homeostatic chemokines (CCL19, CCL21 and CXCL12). Representative of 2 independent repeats (**Video 2**, **Table 3**). (b) Snapshots from a time-lapse movie (**Video 3**) following the interactions of **1G4**-expressing naïve CD8 T-cells (magenta) with antigen-loaded acDCs (green) in a collagen gel containing CXCL12, and the upregulation of CD69 is visible at T = 140 min following the accumulation of an anti-CD69 antibody on the surface of the cells (cyan, also in cyan are irrelevant CD4 T-cells). Representative of 3 independent repeats. (c) Snapshots from a time-lapse movie (**Video 4**) showing three different **1G4**-expressing CD8 T-cells (magenta), loaded with calcium dye Fluo4-AM (green), interacting with an antigen loaded acDC (100 nM NY-ESO-9V, cyan) in 3D collagen and fluxing calcium (green) upon binding. Note the lower flux in the second and third contacts suggesting lower antigen availability, being sequestered by the primary synapse. (d) Speed of **1G4**-expressing naïve CD8 T-cells upon culture in collagen gels containing CXCL12, with acDC either loaded with high affinity (100 nM NY-ESO-9V) or low affinity (100 nM NY-ESO-9L) peptide or unloaded (null). Note the deceleration of the cells upon engaging with high affinity peptide (3 μm/min compared to 12 μm/min for both null and 9L). Representative of 2 independent repeats. (**Videos 7–9**). (e) The cells from (c) extracted from the collagen gel and run on a flow cytometr to look at CD69 and CD25 as activation markers. Note the good activation despite the absence of T-cell arrest with low affinity peptide (NY-ESO-9L). Representative of >3 independent repeats. Scale bars = 10 μm. (****, p<0.0001, ANOVA).

The online version of this article includes the following source data and figure supplement(s) for figure 4:

**Source data 1.** Cell speed in 3D matrices.

**Figure supplement 1.** Motility of T-cells in 3D matrices.

**Figure supplement 2.** Different readouts in 3D.

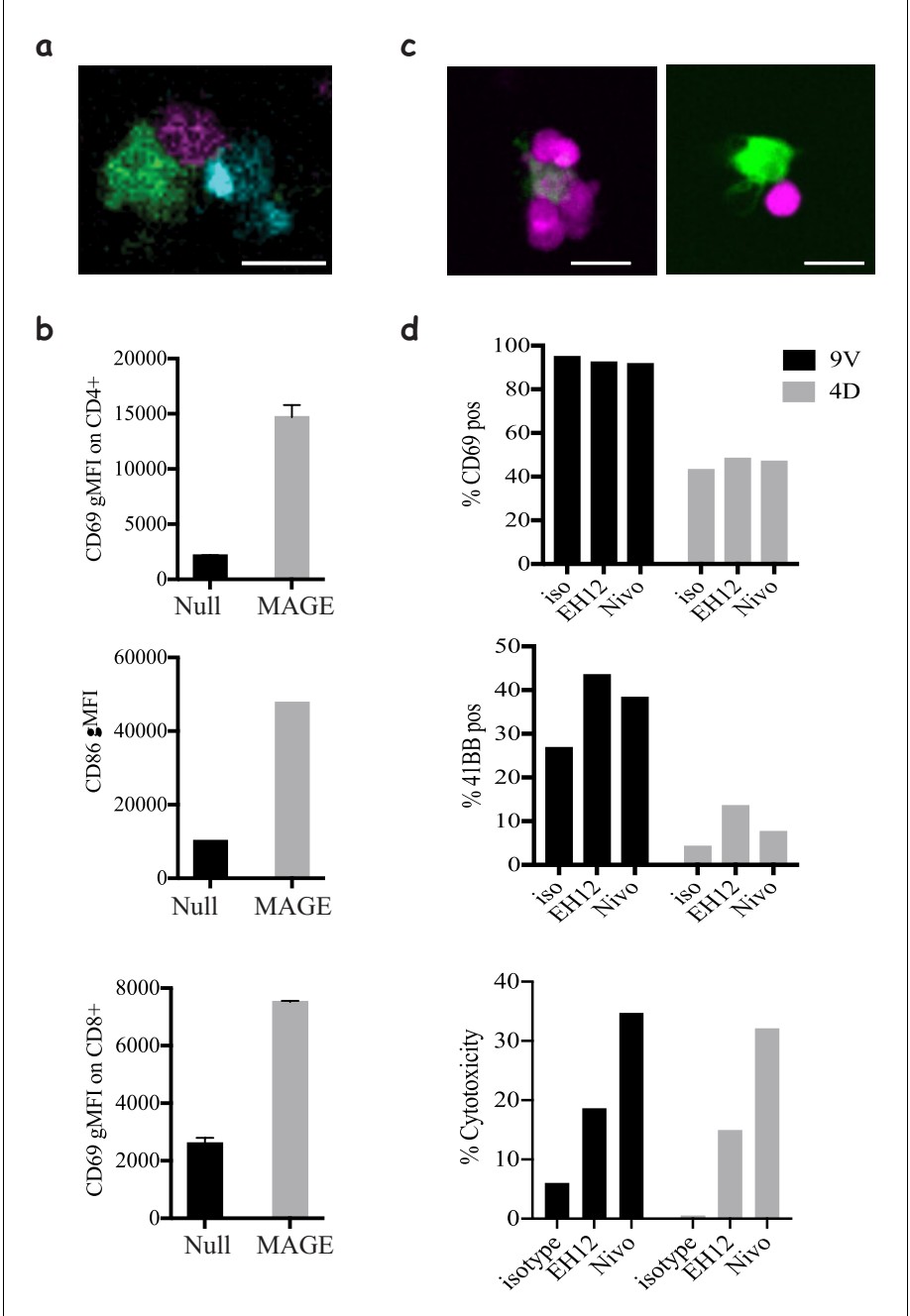

**Figure 5.** Modelling CD4 T-cells' help and immune checkpoint in 3D cultures. (a) Co-culture of **1G4**-expressing naïve CD8 T-cells (magenta), **6F9**-expressing naïve CD4 T-cells (cyan) and acDCs loaded with 100 nM NY-ESO-6T and 10 µM MAGE-A3$_{243-258}$ peptides (green), (***Video 12***), in collagen gels containing CXCL12. (b) Upregulation of CD69 by CD4 (top) and CD8 T-cells (bottom), and the upregulation of the costimulatory molecule CD86 on acDCs (middle) in the co-culture system from (a) in the presence and absence of MAGE as a stimulus for CD4 T-cells. Representative of >3 independent repeats. In all experiments, the cell ratio was 5:5:1 (CD4: CD8: acDC). (c) **1G4**-expressing memory CD8 T-cells (magenta) co-cultured with acDCs loaded with 100 nM NY-ESO-9V (green). Cells form clusters around acDC in the presence of anti PD-1 blocking antibody Nivolumab (left) compared to less tight contacts in the isotype control (right). (d) The activation of the CD8 T-cells by upregulation of CD69, 41BB as well as cytotoxicity measured as the release of LDH following the killing of the target. In all experiments, the cell ratio was 10:1 (CD8: acDC). Scale bars = 10 µm.

The online version of this article includes the following figure supplement(s) for figure 5:

**Figure supplement 1.** PD1 blockade effects on cell motility and regulation.

**Table 3.** Average speeds for the plots in the figures.

| Figure | Condition | Avg. speed [µm/min] ± S.D |
|---|---|---|
| *Figure 4A* | nCD8 no chemokine | 6.41 ± 2.63 |
| | nCD4 no chemokine | 7.02 ± 3.08 |
| | DC no chemokine | 1.01 ± 0.75 |
| | nCD8 CCL19 | 7.83 ± 3.17 |
| | nCD4 CCL19 | 8.7 ± 2.70 |
| | DC CCL19 | 0.99 ± 0.80 |
| | nCD8 CCL21 | 8.07 ± 3.44 |
| | nCD4 CCL21 | 9.38 ± 3.29 |
| | DC CCL21 | 0.97 ± 0.55 |
| | nCD8 CXCL12 | 8.15 ± 2.11 |
| | nCD4 CXCL12 | 8.77 ± 3.02 |
| | DC CXCL12 | 1.16 ± 0.82 |
| | | |
| *Figure 4D* | null | 12.13 ± 2.88 |
| | 9V | 3.88 ± 3.82 |
| | 9L | 12.16 ± 2.81 |
| | | |
| *Figure 2—figure supplement 5A* | Spots | 6.45 ± 1.50 |
| | | |
| *Figure 4—figure supplement 1A* | No chemokine | 1.94 ± 1.80 |
| | CCL19 | 3.466 ± 3.82 |
| | CCL21 | 4.01 ± 2.14 |
| | CXCL12 | 9.27 ± 2.48 |
| | | |
| *Figure 4—figure supplement 1B* | Bovine Collagen I | 7.55 ± 3.48 |
| | Human Collagen I | 7.76 ± 3.10 |
| | Complete Collagen I | 5.94 ± 3.63 |
| | Matrigel | 8.89 ± 3.90 |
| | GelTrex | 7.88 ± 4.00 |
| | | |
| *Figure 4—figure supplement 1C* | Naïve BTX | 10.73 ± 2.96 |
| | Naïve Amaxa U-14 | 8.00 ± 3.80 |
| | Naïve Amaxa T-23 | 6.83 ± 3.85 |
| | Naïve Neon | 9.52 ± 2.20 |
| | Memory BTX | 6.177 ± 2.76 |
| | Memory Amaxa U-14 | 3.391 ± 2.36 |
| | Memory Amaxa T-23 | 2.25 ± 1.89 |
| | Memory Neon | 7.53 ± 4.61 |

## Enhancing CD8 responses through CD4 help and PD1 checkpoint blockade

Our ability to engineer both CD4 and CD8 T-cells independently with different TCRs enables us to interrogate complex dynamics of a trinary cellular system whereby both types of T-cells can interact with similar or different APCs. We show here that using our system we can interrogate the dynamics of CD4 T-cells help to CD8 T-cells (*Castellino et al., 2006*) (*Figure 5A*, *Video 12*) where the

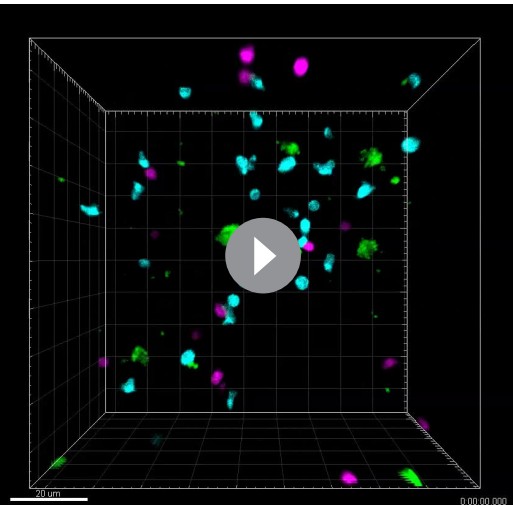

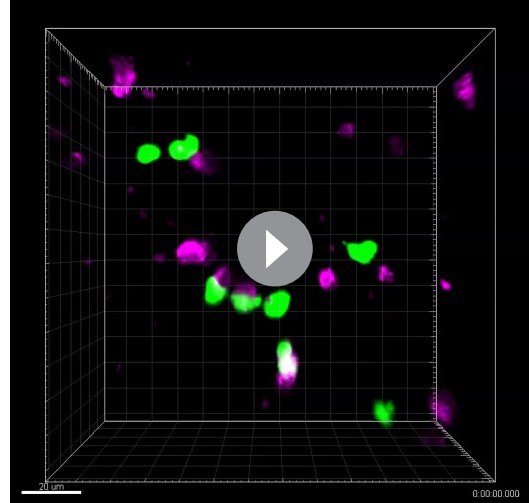

**Video 2.** 3D reconstruction of naïve CD8 (magenta), naïve CD4 (cyan) T-cells, and dendritic cells (green) moving in a 3D collagen gel with RPMI containing human serum and exogenous human CXCL12.
https://elifesciences.org/articles/48221#video2

**Video 3.** 3D reconstruction of **1G4**-expressing naïve CD8 T-cells (magenta), interacting with antigen loaded acDCs (100 nM NY-ESO-9V, green) in 3D collagen in the presence of CCL19. Note the transition of cells between different DCs.
https://elifesciences.org/articles/48221#video3

activation of CD4 T-cells (*Figure 5B*, top) coincides with enhanced maturation of the DCs by upregulation of CD86 (*Figure 5B*, middle) and enhanced activation of the CD8 T-cells (*Figure 5B*, bottom).

Furthermore, we utilised our system to mimic one of the most successful immune checkpoint blockades currently in clinical use; PD-1 blockade,

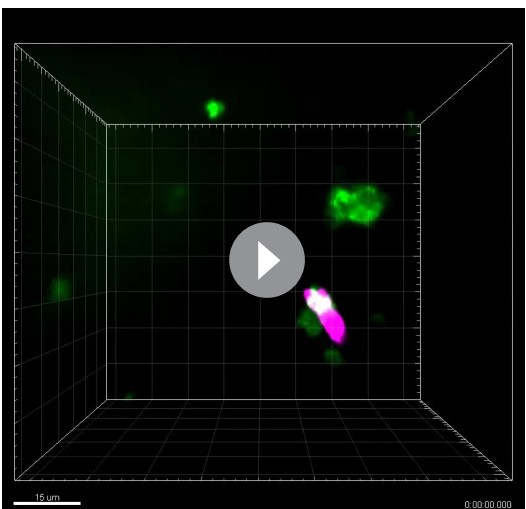

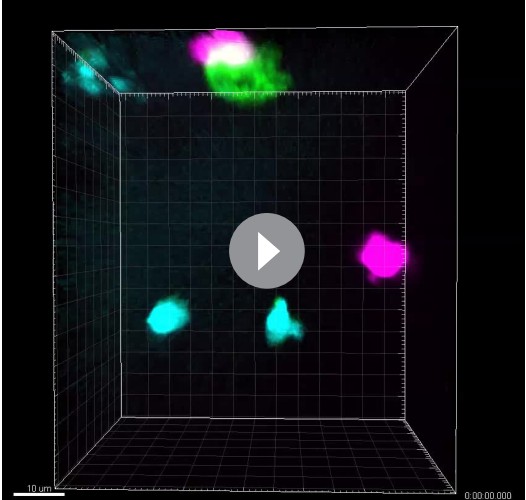

**Video 4.** 3D reconstruction of **1G4**-expressing naïve CD8 T-cells (magenta), interacting with antigen-loaded acDCs (100 nM NY-ESO-9V, green) in a 3D collagen in the presence of CXCL12. Note the intermittent contacts and disengagement of T-cells before re-engaging with a different DC.
https://elifesciences.org/articles/48221#video4

**Video 5.** 3D reconstruction of **1G4**-expressing naïve CD8 T-cells (magenta) interacting with 100 nM NY-ESO-9V loaded acDC (green) in the presence of irrelevant CD4 T-cells (cyan) and a soluble anti-CD69 (cyan). Following conjugate formation, upregulation of CD69 is observed by an accumulation of the antibody (cyan ring) around the CD8 T-cells (magenta).
https://elifesciences.org/articles/48221#video5

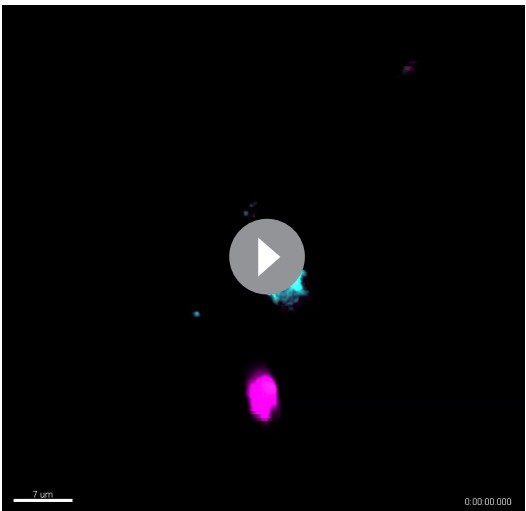

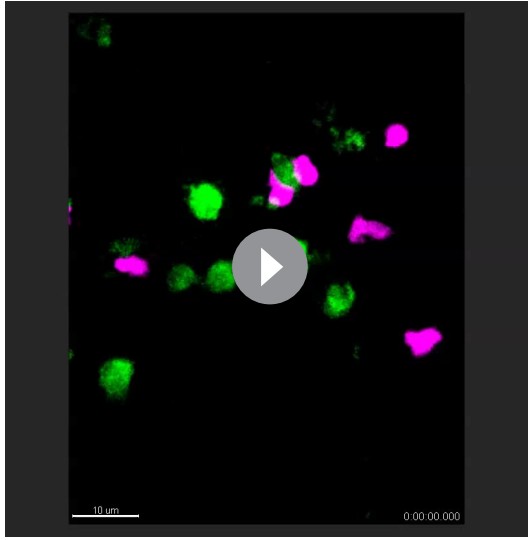

**Video 6.** Maximum projection of 3D time-lapse of 1G4-expressing naïve CD8 T-cells (magenta), loaded with calcium dye Fluo4-AM (green), interacting with antigen loaded acDCs (100 nM NY-ESO-9V, cyan) in a 3D collagen and fluxing calcium (green) upon TCR/pMHC engagement.
https://elifesciences.org/articles/48221#video6

**Video 7.** Maximum projection of 3D time-lapse of 1G4-expressing naïve CD8 T-cells (magenta), interacting with acDCs (100 nM NY-ESO-9V, green).
https://elifesciences.org/articles/48221#video7

which is known to act by enhancing CD8 T-cell responses towards target cells. To that end, we used memory CD8 T-cells and treated them with two clones of blocking antibodies against PD1; Nivolumab and EH12. Despite the lack of any change in the motility dynamics of the cells (*Figure 5—figure supplement 1A*), there was a clear evidence that the presence of the checkpoint blockade enhances the clustering of CD8 T-cells around their targets (*Figure 5C*). This has also been corroborated by enhanced activation, although only observable for late readouts such as 41BB expression and target cell killing (*Figure 5D*). Interestingly, we saw an enhanced expression of PDL-1 (*Figure 5—figure supplement 1B*), the ligand for PD-1, on CD8 T-cells, which could lead to cis interactions that act as a cell intrinsic regulation mechanism (*Sugiura et al., 2019*). Surprisingly, we noted differences between the pathways enhanced by the two blocking antibody clones: EH12 seemed superior at enhancing surface markers, whereas Nivolumab outperformed EH12 in the killing assays.

## Discussion

In this report, we present, to the best of our knowledge, a unique and first of its kind holistic approach to engineer quiescent antigen-specific human T-cells and study them in a context akin to physiological settings. Our tools are widely applicable to generate T-cells expressing different receptors and other proteins with high efficiency and minimal adverse effects associated with other engineering methods, which are essential for implementation of a successful translational approach. We provide a proof of concept where our system is used to probe the dynamics of T-cell/APC interactions as a function of peptide affinity, recapitulate CD4 help to CD8 T-cells and interrogate PD1 checkpoint

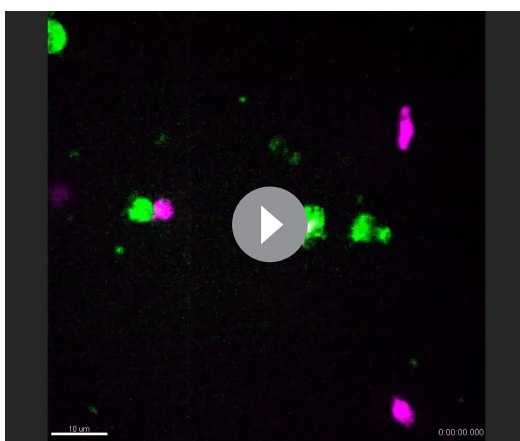

**Video 8.** Maximum projection of 3D time-lapse of 1G4-expressing naïve CD8 T-cells (magenta), interacting with antigen loaded acDC (100 nM NY-ESO-9L, green).
https://elifesciences.org/articles/48221#video8

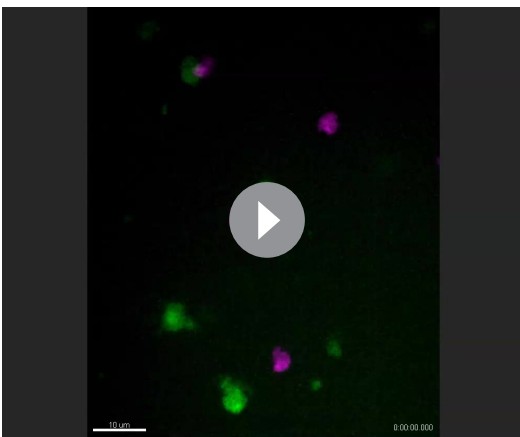

**Video 9.** Maximum projection of 3D time-lapse of **1G4**-expressing naïve CD8 T-cells (magenta), interacting with unloaded acDC.
https://elifesciences.org/articles/48221#video9

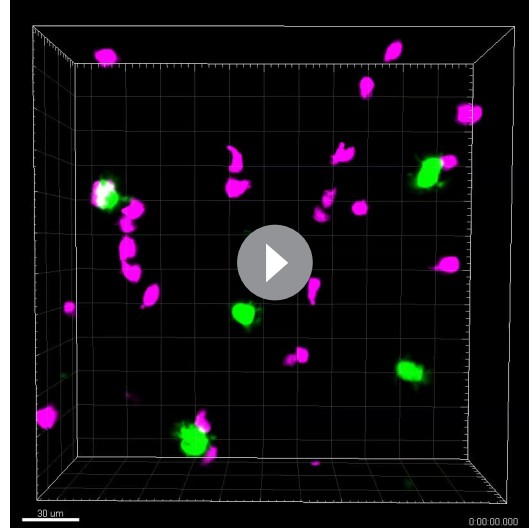

**Video 10.** 3D reconstruction of **1G4**-expressing naïve CD8 T-cells (magenta), interacting with antigen-loaded acDCs (100 nM NY-ESO-6T, green).
https://elifesciences.org/articles/48221#video10

blockade. Our experimental approach can be further extended towards reconstitution of more complex biology such as tumour infiltrating lymphocytes and the role of regulatory T-cells in cancer. We believe that the mechanical and biophysical features introduced by our platform provide greater fidelity to in-vivo conditions and will improve predictive power of pre-clinical studies in all human cell-based systems. We propose that our system should be integrated with other classical and computational approaches to enhance translational research.

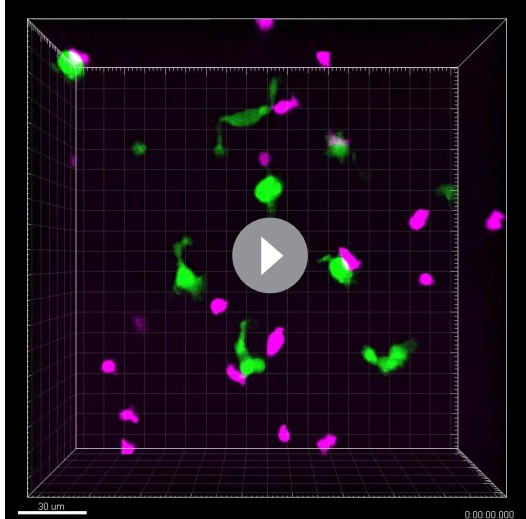

**Video 11.** 3D reconstruction of **1G4**-expressing naïve CD8 T-cells (magenta), interacting with initially unloaded acDCs (green), where the peptide was later added after collagen polymerisation to a final concentration of 100 nM NY-ESO-6T.
https://elifesciences.org/articles/48221#video11

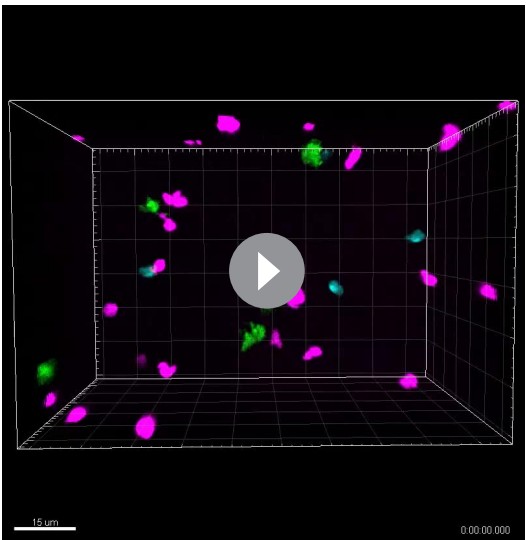

**Video 12.** 3D reconstruction of **1G4**-expressing naïve CD8 T-cells (magenta), **6F9**-expressing naïve CD4 T-cells (cyan) interacting with antigen loaded DCs (100 nM NY-ESO-6T and 10 μM MAGE-A3$_{243-258}$, green).
https://elifesciences.org/articles/48221#video12

## Materials and methods

### Reagents

RPMI 1640 (31870074) was purchased from ThermoFisher Scientific. Hyclone fetal bovine serum (FBS) was obtained from Fisher Scientific. The following anti-human antibodies were purchased from Biolegend: CD62L (DREG-56), CD3 (OKT3), CD8 (HIT8a), HLA-A2 (BB7.2), CD80 (2D10), CD86 (IT2.2), CD40 (5C3), CD69 (FN50), 4-1BB (4B4-1), CD107 (H4A3), TNFα (Mab11) and anti-mouse TCRβ (H57–97). The following anti-human antibodies were purchased from BD Bioscience: CD45RA (HI100), CD4 (RPA-T4), and CD11c (B-ly6), CD14 (MΦP9) and PD-1 (EH12.2H7). Nivolumab (Ab00791-13.12) was obtained from Absolute Antibody. The following anti-human antibodies were purchased from eBioscience: CD83 (HB15e), IFNγ (4S.B3) and PDL-1 (MIH1). HLA-A*0201 with 9V was generated and folded in house. HLA-DPB1*04:01 biotinylated monomers were obtained from the NIH tetramer facility. NY-ESO-9V$_{157-165}$ (SLLMWITQV), NY-ESO-4D$_{157-165}$ (SLLDWITQV), NY-ESO-6T$_{157-165}$ (SLLMWTTQV) and NY-ESO-9L$_{157-165}$ (SLLMWITQL), MAGE-A3$_{243-258}$ (KKLLTQHF VQENYLEY), NY-ESO$_{161-180}$ (WITQCFLPVFLAQPPSGQRA) and HIV p17 GAG$_{77-85}$ (SLYNTVATL) were purchased from GeneScript with >95% purity and resuspended in DMF at 1 mg/ml and stored at −20°C.

### Plasmids and constructs

The sequences for the 1G4 TCR were provided by Vincenzo Cerundolo, the 868 TCR was provided by Andrew Sewell, and all the MHC-II restricted TCRs (SG6, R12C9 and 6F9) were shared by Steven Rosenberg.

All TCR constructs were synthesised using GeneArt services from ThermoFisher; a Kozak sequence was added and codon optimisation for human expression was performed. The **1G4** and **868** constructs included a cysteine modification in the alpha and beta chains (see Supplementary table for sequences).

Constructs for the MHC-II restricted TCRs were synthesised so that the human constant region was replaced with that of mouse (see Supplementary table for sequences).

The alpha and beta chains were cloned separately into a pGEM-GFP-64A plasmid (gift of James Riley) between HindIII and NotI for MHC-I restricted TCRs and for human CD3ζ and between AgeI and NotI for MHC-II restricted TCRs.

### T-cell purification

Human T-cells were isolated from anonymised leukopoiesis products obtained from the NHS at Oxford University Hospitals (REC 11/H0711/7). Resting human T-cell subsets were isolated using negative selection kits (Stemcell Technologies), total CD4 or CD8 T-cells were enriched using RosetteSep kit from Stemcell Technologies, followed by EasySep kit for the corresponding naïve and memory sub-population, following the manufacturer's protocol. Cell purity was assayed with anti-CD4, anti-CD8, anti-CD62L and anti-CD45RA and all cells used had >95% purity.

CD4 and CD8 T-cells were cultured at 37°C, 5% CO$_2$, in RPMI 1640 (Roswell Park Memorial Institute) medium supplemented with 10% FBS (Gibco), 5% penicillin-streptomycin (PenStrep, Gibco), 1x MEM non-essential emino acids solution, 20 mM HEPES, 1 mM sodium pyruvate, 2 mM Glutamax and 50 µM 2-mercaptoethanol (Sigma) (all from Thermo Fisher unless stated otherwise).

For T-cell activation, 1:1 anti-CD3/anti-CD28- coated T-cell activation beads (11132D) were added at a 1:1 ratio with 50 U/ml of IL-2 for two days, followed by removal of the beads and propagation of the culture in IL-2 until day 7.

### Express monocyte-derived dendritic cell generation

Monocytes were enriched from the same leukopoiesis products as T-cells using RosetteSep kit (Stemcell Technologies), and were then cultured at 1−2 × 10$^6$/ml in 12-well plates with 1 ml of differentiation medium containing 50 ng/ml Interleukin 4 (IL-4, 200–04 A, Peprotech) and 100 ng/ml granulocyte-monocyte colony stimulating factor (GM-CSF, 11343125, Immunotools) for 24 hr. For maturation the following cytokines were added for an additional 24 hr: 1 µM prostaglandin E$_2$ (PGE$_2$, P6532, Sigma), 10 ng/ml interleukin one beta (IL1β, 201-LB-025/CF, Biotechne), 50 ng/ml tumour necrosis factor alpha (TNFα, 300-01A, Peprotech) and 20 ng/ml interferon gamma (IFNγ,

285-IF-100/CF, Bio-techne). Cell purity of the monocyte population was assessed using antibodies against CD14 and CD11c and typically was above 80%. Any non-monocyte contaminants should be removed by adhering the cells for 2–3 hr immediately after isolation followed by washing off any unbound cells and applying the differentiation media.

Peptide loading on DC was quantified using an AlexaFluor 647 conjugated high affinity soluble TCR (c113 against NY-ESO) (*Zhao et al., 2007*) with a known dye ratio and Quantum MESF Alexa-Fluor 647 beads as calibration (647, Bangs Laboratories).

Secretion of soluble factors was assessed using 0.5 ml of culture medium, and the corresponding controls of media containing the differentiation and activation cytokines, following the manufacturer's protocol (Proteome Profiler Human XL Cytokine Array kit, ARY022B, R &D Systems).

## HLA typing of donors

For antigen-specific experiments, the donors were assessed for the relevant HLA. For CD8-based experiments, 50 µl of whole blood was taken for flow cytometry analysis using an anti-HLA-A2 mAb (clone BB7.2). For CD4 -based experiments, 50 µl of blood was used to extract DNA with the QIAamp DNA Mini kit (51304, Qiagen). The extracted DNA was used for PCR analysis using the All-Set+ Gold HLA DPB1 High-Resolution Kit (54070D, VH Bio) to determine whether or not the suitable haplotype was expressed. We were also able to induce the expression of a single chain dimer of HLA-A*02 in moDCs from donors which were HLA-A*02-negative donors using the mRNA electroporation approach.

## Engineering antigen-specific T-cells

To express TCR constructs within resting naïve and memory T-cells, we used mRNA electroporation. mRNA for the relevant TCRα, TCRβ and CD3ζ chains was prepared from a linearised pGEM-64A vector or a T7 containing PCR product was done using mMESSAGE mMACHINE T7 ULTRA Transcription kit as per manufacturer's protocol (ThermoFisher, AM1345). The mRNA was purified using MegaClear kit (ThermoFisher, AM1908) and was aliquoted and kept at −80˚C (*Zhao et al., 2006b*). To achieve high efficiency of expression, mRNA quality was assessed using agarose gels and samples showing any sign of degradation were discarded. Any mRNA preparation with yields bellow 40 µg mRNA per 1 µg of DNA was considered of low quality; finally, all mRNA aliquots were stored at concentrations > 1 µg/µl, which was achieved by concentrating the mRNA product using ammonium acetate precipitation or by combining >3 reactions of in vitro transcription during the MegaClear clean-up step.

T-cells were harvested and washed three times with Opti-MEM (LifeTechnologies). The cells were resuspended at $25 \times 10^6$/ml and $2.5-5 \times 10^6$ cells were mixed with the desired mRNA products and aliquoted into 100–200 µl per electroporation cuvette (Cuvette Plus, 2 mm gap, BTX). For $10^6$ cells CD8 T-cells, 2 µg of each TCRα, TCRβ and CD3ζ RNA was used. For $10^6$ CD4 T-cells, 4 µg of TCRα and TCRβ was used. Cells were electroporated at 300 V, 2 ms in an ECM 830 Square Wave Electroporation System (BTX). The cells were then collected from the cuvette and cultured in 1 ml pre-warmed media. Amaxa and Neon electroporation systems were also tested but failed in either achieving efficient TCR expression or affected T-cell motility. For Amaxa electroporation, the human T-cell nucleofector kit (VAPA-1002) was used and the manufacturer's protocol was followed. Three different settings were tested: V-24, U-14 and T-23. For Neon electroporation, $2.5 \times 10^6$ cells were resuspended in a 100 µl tip, and two settings were tested (1700 V 10 ms, 4 pulses and 2150 V, 20 ms one pulse) (*Aksoy et al., 2019*).

Note, extended culture (>5 days) prior to electroporation also results in marginal reduction in TCR expression efficiency, although no reduction is observed for control GFP.

The exogenous TCR can be detected up to 96 hr post electroporation.

## Supported lipid bilayers (SLB)

SLBs were prepared as previously described (*Dustin et al., 2007*). In brief, piranha and plasma cleaned coverslips were mounted on sticky-Slide VI0.4 chamber (ibidi). Small unilamellar liposomes (LUVs) were prepared using 4 mM 18:1 DGS-NTA(Ni), 'NTA-lipids', (790404C-AVL, Avanti Polar Lipids), 4 mM CapBio 18:1 Biotinyl Cap PE (870282C-AVL, Avanti Polar Lipids), 'CapBio-lipids', and 4 mM 18:1 (Δ9-Cis) PC, 'DOPC-lipids', (850375C-AVL, Avanti Polar Lipids,). NTA-lipids were used at a

final concentration of 2 mM and CapBio lipids were used at a dilution resulting in site density of 100 molecules/μm (*Council MRC, 2008*). SLBs were allowed to form by incubating the coverslips with the appropriate LUVs for 20 min at room temperature (RT) followed by a wash with HEPES-buffered saline (HBS) supplemented with 1 mM $CaCl_2$ and 2 mM $MgCl_2$, and human serum albumin (HBS/HSA). The SLBs were loaded with saturating amounts of AlexaFluor568 labelled streptavidin, 5 μg/ml (S11226, ThermoFisher) for 10 min at RT followed by pMHC at 100 molecules/μm (*Council MRC, 2008*) and ICAM-1 200 molecules/μm (*Council MRC, 2008*) for 20 min at RT.

## Micro-contact printed chambers

Micro-patterned surfaces presenting pMHC molecules were prepared using micro-contact printing by modifying the procedures described previously (*Mayya et al., 2018*). Topological masters with repetitive circle patterns of 10 μm in diameter, spaced 30 μm centre-to-centre on a square grid were used to cover the entire channel of a sticky-Slide $VI^{0.4}$ chamber (ibidi). The master was used for casting of polydimethylsiloxane (PDMS) elastomer stamps. Sylgard 184 (Dow Corning) was used by mixing 1/7 (v/v) of curing agent to the elastomer. Rectangular stamps of PDMS were coated with biotinylated, AlexaFluor 674 labelled Fc IgG (AG714, Merck Millipore) at 2 μg/ml in 150 μl of PBS for one hour. The blocks were then rinsed extensively in PBS, PBS with 0.05% Tween-20 and finally in MilliQ-grade water followed by gentle drying with $N_2$ to remove droplets of water. Fc coated PDMS blocks were stamped onto the cleaned coverslips for 5 min under ~20 g of load for efficient transfer. Attempts to directly stamp pMHC, avidin or streptavidin resulted in partially or completely non-functional proteins. Streptavidin stamps were also non-uniform and eroded with subsequent washes. The patterned coverslip was then affixed to the sticky-Slide $VI^{0.4}$ and washed sequentially with MilliQ-grade water and PBS, then coated with 13.5 μg/ml of CCL21 in 30 μl for one hour, followed by 3 μg/ml of ICAM1 in 180 μl for 3 hr at 37°C. Coverslips were then used immediately for the following steps or kept in PBS at 4°C overnight. The stamped and protein coated channels were blocked with 10% dialysed FCS (to remove any free biotin) for 30 min at RT. Then unlabelled streptavidin was introduced into the channel at 2 μg/ml for 10 min at RT, followed by washing and introduction of 9V/A2 pMHC at 1 μg/ml for 20 min at RT.

## Collagen gel chamber assembly

DCs were loaded with the peptide specified in the figures for 90 min at 37°C. For imaging experiments, T-cells were labelled with a volumetric dye as described below (*Imaging*) or loaded with cell trace violet (C34557, ThermoFisher) for proliferation assays, or otherwise kept unlabelled.

Collagen mix was made by modifying the approach of *Gunzer et al. (2000)* using 5 μl 7.5% sodium bicarbonate, 10 μl 10x MEM, 75 μl 3 mg/ml Bovine Collagen I (PureCol) and a chemokine, 1 μg/ml of CCL21 (300-35-100, Peprotech) or CCL19 (300-29B, Peprotech) or 0.3 μg/ml of SDF1-a/CXCL12 (300-28A, Peprotech). All the work with collagen was done on ice to prevent polymerisation. Other matrices tested: VitroCol Human Collagen I (5007-A, CellSystems), Matrigel (at final conc. of ~4 mg/ml, two batches tested with similar results), and GelTrex (A1413301, ThermoFisher, at final conc.n of ~8 mg/ml, two batches tested with similar results). All 3D cultures were made with bovine collagen I, except for the comparison shown in *Figure 4—figure supplement 1B*, and mainly with CXCL12 unless otherwise stated.

Cell-containing media (15 μl) supplemented with 10% Human serum (S-101B-FR, APS) (or 10% FBS) wasadded to 35 μl of the collagen mix so that the final culture contained 200,000 CD8, 200,000 CD4s T-cells and DCs at varying ratios (typically 1:1, 1:5 and 1:10 to T-cells). Lower cell numbers can be used for functional experiment but the reported numbers here are ideal for good density for imaging.

Collagen mix is then added to a μ-slide angiogenesis chamber or sticky-Slide $VI^{0.4}$ chamber (ibidi) and put upside down in a 37°C incubator with 5% $CO_2$ for ~60 min (to entrap the cells in the gel as it polymerises). The wells were then topped up with 30 μl media containing the same concentration of chemokines as the gel (for μ-slide angiogenesis) or 100 μl per well (for sticky-Slide $VI^{0.4}$). For in gel loading experiments, the media contained a peptide at 2x the reported concentration. Samples were either taken for imaging or kept in the incubator for functional analysis. For PD-1 blockade, the antibody or isotype control was added to the T-cells for 15 min prior to incorporation in the gel so the final antibody concentration was 10 μg/ml.

Extended culture in the 3D system leads to ~50% reduction in cell speed due to chemokine desensitisation and can be overcome by the addition of fresh chemokine after 18 hr.

## Flow cytometry

**1G4** TCR expression was assessed using 9V-HLA-A2 tetramers, **868** TCR expression was assessed using SL9-HLA-A2 streptamers (6-7004-001, iba) and **6F9**, **SG6** and **R12C9** expression was assessed with anti-mouse TCR β (H57) antibody. The MHC II restricted TCRs seem to have very low affinity towards their antigen, preventing detection using tetramers (attempts to measure affinity using SPR suggest a $K_D$ above 500 μM). In brief, 50,000 cells were stained with 2.5 μg/ml tetramers (9V/A2) or 5 μg/ml streptamers (SL9/A2) or 1 μg/ml antibody for 20 min at 4°C, washed with PBS containing 2% FBS and 2 mM EDTA and taken for analysis on a flow cytometr (FORTESSA X-20, BD Biosciences).

To analyse samples from collagen gels, the gels were first digested into solution using collagenase VII (Sigma #C0773) at 100 U/ml for 1 hr at 37°C. The supernatant was subsequently removed, and the cells were stained with the desired antibody combination for 20 min 4°C. For intracellular staining, the cells were cultured in collagen gels and brefeldin A or monensin was added, after 20 hr for 2 hr to prevent cytokine secretion to the supernatant and retain them inside the cells, along with stimulation of phorbol 12-myristate 13-acetate (PMA, P8139, Sigma) at 1 μg/ml and ionomycin (I0634, Sigma) at 1 μM, the cells were then stained using the FoxP3 intracellular staining kit (00-5523-00, eBioscience) and the desired antibodies. For experiments measuring LAMP1 expression, an antibody against it was added at the start of the culture.

LDH Cytotoxicity Detection kit (MK401, Takara Bio) was used as per manufacturer instructions to detect cell killing.

## Imaging

Cells were labelled with one of the volumetric dyes (all from ThermoFisher): CMFDA (250 nM) ), CMRA (500 nM) or Deep Red (100 nM) for 20 min in complete media. For imaging CD69 activation, the collagen mixture described above was supplemented with a BV421 labelled anti-CD69 at 1 μg/ml (adjustments in volume are made to the media to maintain the same final collagen concentration). For calcium imaging, T-cells were loaded with 1 μM of Fluo4-AM (ThermoFisher) for 20 min at 37°C. Cells were imaged using either a Dragonfly Spinning Disk system, a Perkin Elmer Spinning disk or an Olympus FluoView FV1200 confocal microscope using a 30x Super Apochromat silicone oil immersion objective with 1.05 NA. All microscopes included an environmental chamber to maintain temperature at 37°C and $CO_2$ at 5%. Time-lapse images were acquired every 30 or 60 s. z-Scans were acquired every 3 μm.

SLB imaging was performed on an Olympus IX83 inverted microscope equipped with a TIRF module and Photomertrics Evolve delta EMCCD camera using an Olympus UApON 150x TIRF N.A 1.45 objective.

## Data analysis

Microscopy data from collagen gels was rendered and analysed using built-in tools in IMARIS software (Bitplane). Speeds bellow 2 μm/min were considered to correspond to stationary cells. Synapse images were analysed using Fiji. Image analysis for micro-contact printing data was done using TIAM (*Mayya et al., 2015*) as described in *Mayya et al. (2018)*. In brief, cells were tracked using DIC images. IRM and fluorescence signal-positive segments of the tracks were used to calculate the attachment rate and the number of cells interacting with stimulatory spots to extract the half-life of interactions where a simple first order rate was used dA/dt=-k[A]; where A is cell number and k is the off-rate constant representing the exit from the spots by dissolution of the IS. Data from IMARIS and FlowJo were plotted and analysed in Prism (GraphPad), where all statistical tests were performed.

## Acknowledgements

We thank Vincenzo Cerundulo for the **1G4** TCR sequence, Andrew Sewell for the **868** TCR plasmid, Steven Rosenberg for the **6F9**, **R12C9** and **SG6** TCR sequences and James Riley for pGEM-64A plasmid and introducing us to mRNA electroporation. We thank Jehan Afrose and Heather Rada for the generation of recombinant proteins. We thank the NIH tetramer facility for providing pMHC

monomers. We thank Arbel Artzy-Schnirman for critical reading of the manuscript and help with graphical figures. We thank Kinneret Keren, Erez Braun, Anton van der Merwe, Marion H Brown and David Depoil for discussions and feedback.

The work has been supported by a UCB-Oxford Post-doctoral Fellowship to EA-S, European Research Council ERC-2014-AdG_670930 for VM and SB, Kennedy Trust for Rheumatology (KTRR) Prize Studentship to PD, a Wellcome Trust Principal Research Fellowship 100262Z/12/Z, a grant from KTRR and Human Frontiers Science Program Research Grant RGP0033/2015 to MLD, and a Wellcome Trust Senior Research Fellowship (207537/Z/17/Z) for OD.

## Additional information

### Competing interests

Michael L Dustin: Reviewing editor, *eLife*. The other authors declare that no competing interests exist.

### Funding

| Funder | Grant reference number | Author |
|---|---|---|
| Wellcome Trust | 100262Z/12/Z | Michael L Dustin |
| UCB UK | UCB-Oxford AZR00960 | Enas Abu-Shah |
| Wellcome Trust | 207537/Z/17/Z | Omer Dushek |
| European Research Council | ERC-2014-AdG_670930 | Štefan Bálint Viveka Mayya |
| Human Frontier Science Program | RGP0033/2015 | Michael L Dustin |
| Kennedy Trust for Rheumatology | | Philippos Demetriou |

The funders had no role in study design, data collection and interpretation, or the decision to submit the work for publication.

### Author contributions

Enas Abu-Shah, Conceptualization, Resources, Data curation, Formal analysis, Funding acquisition, Validation, Investigation, Methodology, Writing—original draft, Project administration, Writing—review and editing; Philippos Demetriou, Methodology, Writing—review and editing; Štefan Bálint, Data curation, Methodology, Writing—review and editing; Viveka Mayya, Data curation, Software, Formal analysis, Methodology, Writing—review and editing; Mikhail A Kutuzov, Resources, Writing—review and editing; Omer Dushek, Michael L Dustin, Conceptualization, Supervision, Funding acquisition, Writing—review and editing

### Author ORCIDs

Enas Abu-Shah https://orcid.org/0000-0001-5033-8171
Štefan Bálint http://orcid.org/0000-0003-4470-5881
Viveka Mayya https://orcid.org/0000-0001-5668-9124
Omer Dushek https://orcid.org/0000-0001-5847-5226
Michael L Dustin https://orcid.org/0000-0003-4983-6389

### Ethics

Human subjects: This project has been approved by the Medical Sciences Inter-Divisional Research Ethics Committee of the University of Oxford REC 11/H0711/7 to cover the use of human blood products purchased from National Health Services Blood and Transplant service (NHS England).

Decision letter and Author response
Decision letter https://doi.org/10.7554/eLife.48221.sa1
Author response https://doi.org/10.7554/eLife.48221.sa2

## Additional files

### Supplementary files
• Transparent reporting form

### Data availability
No new gene datasets were generated during this study. Source data files have been provided for Figures 2, 3, and 4. The TCR sequences used have been published in the past in the literature (cited in the manuscript) and the modifications made are clearly stated in the tables in the manuscript. The constructs are available through a request to the corresponding authors of the previously published articles.

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
