## [Decision Letter]

[Editors’ note: this article was originally rejected after discussions between the reviewers, but the authors were invited to resubmit after an appeal against the decision.]

Thank you for submitting your work entitled "A tissue-like platform for studying engineered quiescent human T-cells' interactions with dendritic cells" for consideration by *eLife*. Your article has been reviewed by three peer reviewers, and the evaluation has been overseen by a Reviewing Editor and a Senior Editor. The reviewers have opted to remain anonymous.

Our decision has been reached after consultation between the reviewers. Based on these discussions and the individual reviews below, we regret to inform you that your work will not be considered further for publication in *eLife*.

Reviewer #1:

This is a well written, clear report that describes a technical advance to study antigen-specific interactions between CD4 and CD8 T cells and dendritic cells. The overall concept is of technical interest, but as it stands the manuscript does not report any new mechanistic insight into T cell-APC interactions. It is not clear to me that this is suitable for *eLife* and may be better suited to a methods journal.

1) One of the reported benefits is the ability to electroporate naive or memory T cells, but as far as I can tell only data from naive T cells are reported here – what happens with memory T cells?

2) A lot of the relevant data seem to be in the supplementary data, making it overall hard to follow what are the most important experiments. More of the supplementary data should be in the main manuscript.

Reviewer #2:

The authors here demonstrated their own technique for engineering the human T cell and 3D culture system for evaluating the interaction between engineered human T cells and DCs. The method would be of some interest in the field of human immunology for breaking the current experimental limitation in this field, although the presented results seem to have several serious problems.

The authors mentioned that the square- wave electroporator could maintain the motility of T cells, and also showed the usage of collagen-based 3D matrix resulted in better basal motility of cells. However, it cannot be evaluate these data without appropriate controls. I would recommend the data of cell motility would be presented as comparison between the group using general electroporation and that with the square-wave electroporator. In addition, the authors claim the cell motility is maintained in 3D culture system, but nobody knows if it is similar or different to those seen in vivo LN tissue. The authors could compare the difference of cell motility in 3D culture system and in vivo using mouse model such as transferred human T cells into SCID mice.

Reviewer #3:

The authors present a novel pipeline for the generation of TCR-transgenic human CD4 and CD8 T cells as well as dendritic cells from the same donor. The system relies on square-wave electroporation of primary human CD4 and CD8 T cells with mRNAs coding for optimized TCR α/β (having either a novel Cys-Cys bridge for linking both chains in CD8 cells or a murine transmembrane domain for CD4 cells) and human CD3 Zeta chains. They demonstrate robust expression of a variety of antiviral or anti-tumor TCRs in both, CD4 and CD8 cells. Key point is the fact, that the motility of the T cells is not hampered by the transgenic approach and hence allows to study T cell activation processes by life cell imaging in artificial 3-D collagen matrices.

The authors are to be complimented on getting such a complicated system going. However, despite some minor technical issues I also miss the application of this approach for the identification of a new biological insight. Specific comments are below:

The key criticism is that there is essentially only a demonstration of the functionality of the technology. No attempts are being made to find something new in terms of T cell biology by its use. E.g. by the peptide adding assay it could be measured, how long it takes from application until the first measurable response. Modulators could be added to see the effect e.g. of checkpoint blockers or the like etc.. Anything is welcome, but I would really want to see the method applied to a biological question. Currently the manuscript appears a bit uninspired due to this shortcoming.

---

## [Author Response]

Two major points were made by the reviewers:

1) The format of the paper and its suitability for *eLife*. We would like to emphasise that we have submitted this manuscript as a methods paper under the Tools and Resources section of *eLife*. However, the reviewers’ comments indicate concerns that this seems like a methodological paper and lacks new biological insight. We would like to highlight that this is indeed meant to be a methodological advancement that is validated by reproducing known biological results. As part of establishing our system, we verified it reproduces known immunological processes, such as CD4 help to CD8 T-cells (Figure 5). To further strengthen the utility of our system, we now also include data using anti-PD1 blockade (see below) showing enhanced CD8 responses. To avoid confusion on the paper’s format, we have included changes to the text to highlight and clarify the aim of the paper.

2) The structure of the paper, in particular the separation between main figures and supplementary data. We have now re-organised and extended the main figures while maintaining some data in supplementary figures. We are hoping to take advantage of the unique format of *eLife* where the supplementary data are clearly entrained to the main figures, making the presentation a continuous flow and not as strictly separated as traditionally done in other journals where making the link between primary and supplementary figures is not so clear.

As we have argued, and as the reviewers agree, there is an urgent need to establish simple and robust ways to manipulate human T-cells to facilitate translational research in human immunology. This has been a concern for many researchers working in the field and we believe that our tool will be of great value to the community.

Reviewer #1:

This is a well written, clear report that describes a technical advance to study antigen-specific interactions between CD4 and CD8 T cells and dendritic cells. The overall concept is of technical interest, but as it stands the manuscript does not report any new mechanistic insight into T cell-APC interactions. It is not clear to me that this is suitable for eLife and may be better suited to a methods journal.

We thank the reviewer for appreciating the presentation and importance of our work. This paper is intended to be a methodological paper and was submitted to *eLife* under the Tools and Resources section, which is equivalent to a methods section. We do believe that a more widely accessible journal like *eLife* will allow for the dissemination of our findings to a larger community including immunologist as well as cancer researchers.

1) One of the reported benefits is the ability to electroporate naive or memory T cells, but as far as I can tell only data from naive T cells are reported here – what happens with memory T cells?

To address this comment, we now included measurements of memory T-cells’ responses (Figure 2—figure supplement 2A). We have also performed the experiments on PD-1 checkpoint blockade with memory CD8 T-cells as naïve cells do not express this protein (Figure 5, and response to reviewer 3). [subsection “Enhancing CD8 responses through CD4 help and PD1 checkpoint blockade”]. In addition, we have used the cells electroporated with the TCR, both memory and naïve, on supported lipid bilayers in a different manuscript currently on Biorxiv (Demetriou et al., bioRxiv 589440).

2) A lot of the relevant data seem to be in the supplementary data, making it overall hard to follow what are the most important experiments. More of the supplementary data should be in the main manuscript.

We have now re-organised the figures.

Reviewer #2:

The authors here demonstrated their own technique for engineering the human T cell and 3D culture system for evaluating the interaction between engineered human T cells and DCs. The method would be of some interest in the field of human immunology for breaking the current experimental limitation in this field, although the presented results seem to have several serious problems.

*The authors mentioned that the square- wave electroporator could maintain the motility of T cells, and also showed the usage of collagen-based 3D matrix resulted in better basal motility of cells. However, it cannot be evaluate these data without appropriate controls. I would recommend the data of cell motility would be presented as comparison between the group using general electroporation and that with the square-wave electroporator. In addition, the authors claim the cell motility is maintained in 3D culture system, but nobody knows if it is similar or different to those seen* in vivo *LN tissue. The authors could compare the difference of cell motility in 3D culture system and* in vivo *using mouse model such as transferred human T cells into SCID mice.*

We have added as part of the supplementary data results highlighting the differences between three electroporation methods: Neon by ThermoFisher, Amaxa by Lonza and the BTX square-wave electroporator, with analysis of expression, viability and recover of cells (Figure 2—figure supplement 3) and data regarding motility in the collagen gels and how it reduced activation (Figure 4—figure supplement 1C-F).

As can be seen the recovery using the other methods is poor, in particular the Neon system does not seem to be resulting in satisfactory expression levels and the Amaxa results in cell toxicity evident in increased stationary cell numbers and lower speed (in particular in the memory T-cell population). [subsection “Collagen-based 3D model to support immune cell migration and interactions”].

We thank the reviewer for pointing out that it would be helpful to confirm that our results are consistent with dynamics of human T-cells in tissue settings. We have now added reference to two such studies. Human T-cells have been tracked in lymph nodes of immunodeficient mice and the rate of movement is very similar to what we observe here (Murooka et al., 2012). In addition, human T-cells have also been tracked in explanted tissues in which labeled T-cells were applied and allowed to invade. Again, the movement parameters are very similar to what we have measured here (Salmon et al., 2012 and Bougherara et al., 2015). The robustness of these systems is likely due to the switch like response of T-cells to the chemokinetic signals, which are fully activated above a threshold (2007).

Reviewer #3:

[…] The authors are to be complimented on getting such a complicated system going. However, despite some minor technical issues I also miss the application of this approach for the identification of a new biological insight. Specific comments are below:The key criticism is that there is essentially only a demonstration of the functionality of the technology. No attempts are being made to find something new in terms of T cell biology by its use. E.g. by the peptide adding assay it could be measured, how long it takes from application until the first measurable response. Modulators could be added to see the effect e.g. of checkpoint blockers or the like etc.. Anything is welcome, but I would really want to see the method applied to a biological question. Currently the manuscript appears a bit uninspired due to this shortcoming.

We thank the reviewer for appreciating the complexity of the system and appreciating its usefulness. As this has been constructed as a methods paper with the aim of making it widely accessible to the research community without further delay, we do not explicitly try to provide a new biological finding. We however opted to reconstituting known immunological responses to establish its validity (e.g. CD4 help to CD8s). However, following the reviewer’s comment we have now included data showing anti-PD1 blockade enhances memory CD8 responses and also observed the upregulation of PDL1 on CD8s in these settings which can be in harmony with data suggesting the existence of a cis interactions between PD1/PDL1 as a mechanism of regulating this inhibitory receptor, furthermore, we identify that the blockade contributes to increasing cell clustering (Figure 5B).

We see in our data and some of the videos presented as part of this manuscript (specifically CD69 upregulation) that we can detect responses as early as couple of hours from the start of imaging (which is ~3hrs from combining the T-cells with peptide loaded DC in the collagen gels).